# Nucleophilic aromatization of monoterpenes from isoprene under nickel/iodine cascade catalysis

Wei-Song Zhang[1,2], Ding-Wei Ji[1], Yang Yang[1], Ting-Ting Song[1], Gong Zhang[1,2], Xiao-Yu Wang[1,2] & Qing-An Chen [1,2] ✉

As a large number of organic compounds possessing two isoprene units, monoterpenes and monoterpenoids play important roles in pharmaceutical, cosmetic, agricultural, and food industries. In nature, monoterpenes are constructed from geranyl pyrophosphate (C10) via various transformations. Herein, the bulk C5 chemical—isoprene, is used for the creation of various monoterpenoids via a nucleophilic aromatization of monoterpenes under cascade catalysis of nickel and iodine. Drugs and oil mixtures from conifer and lemon can be convergently transformed to the desired monoterpenoid. Preliminary mechanistic studies are conducted to get insights about reaction pathway. Two types of cyclic monoterpenes can be respectively introduced onto two similar heterocycles via orthogonal C–H functionalization. And various hybrid terpenyl indoles are programmatically assembled from abundant C5 or C10 blocks. This work not only contributes a high chemo-, regio-, and redox-selective transformation of isoprene, but also provides a complementary approach for the creation of unnatural monoterpenoids.

Owing to their unique pharmacological properties, such as antimicrobial, anti-inflammatory, antipruritic and analgesic activities[1–4], monoterpenes and monoterpenoids have been widely applied in the clinical therapeutics and medicinal discovery. Besides, these compounds also play important roles in cosmetic, food and perfume industries[5–8]. In nature, isopentenyl pyrophosphate (IPP) and dimethylallyl pyrophosphate (DMAPP) are condensed to form the geranyl diphosphate (GPP) by geranyl pyrophosphate synthase (GPPase). Through a key intermediate (geranyl cation), cyclic or acyclic monoterpenes are synthesized from GPP under enzymatic catalysis[9–11]. Different monoterpenes could be further modified by other specific enzymes for the construction of various functionalized monoterpenoids (Fig. 1a). Considering the importance of terpenoids, it has broad prospects and great value to develop selective synthesis of terpenoids from bulk chemicals by mimicking the biological process.

Given its characteristic structure and abundance, isoprene is undoubtedly the most cost-effective choice for biomimetic constructions of terpenoids. In terms of the synthesis of hemiterpenoids, various catalytic systems have been well developed under transition metal catalysis[12–35]. Besides, with the regulation of ligand, Beller, Finn, Réau, Navarro, Carbó and our group reported palladium catalyzed nucleophilic telomerization of isoprene to produce acyclic monoterpenoids[32,36–44]. Using nickel-aminophosphinite complexes, different dimers of isoprene mixed together with low yields in Mortreux's work[45]. With the aid of N-heterocyclic carbene (NHC) ligands, we have recently demonstrated a selective heteroarylative cyclotelomerization of isoprene for the construction of cyclic monoterpenoids (Fig. 1b)[35,46]. Given aromatic compounds play indispensable roles in a broad spectrum of industries, we wonder is it possible to develop a strategy for the creation of aromatic monoterpenoids with a distinctive selectivity. To realize this proposal, we had to address the following challenges: (1) Chemoselectivity. The existing literatures have demonstrated that the formation of hemiterpenoids (C5), acyclic monoterpenoids (C10) was more favorable over cyclic

[1]Dalian Institute of Chemical Physics, Chinese Academy of Sciences, Dalian 116023, People's Republic of China. [2]University of Chinese Academy of Sciences, Beijing 100049, People's Republic of China. ✉e-mail: qachen@dicp.ac.cn

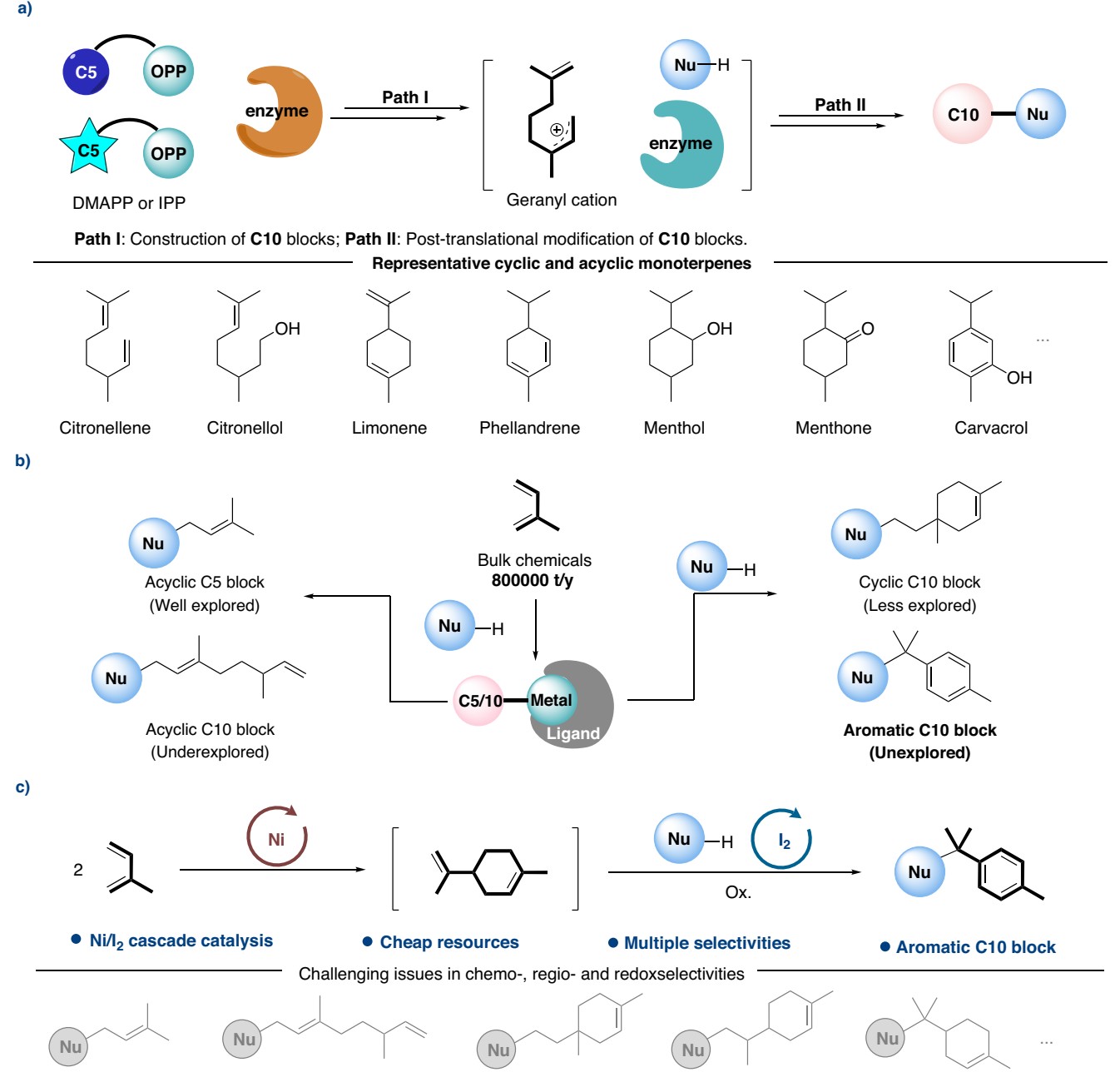

**Fig. 1 | Catalytic construction of terpenoids. a** Biosynthetic pathway for the construction and post-translational modification of C10 blocks; **b** Biomimetic nucleophilic reactions with isoprene under transition metal catalysis; **c** This work: Nucleophilic aromatization of monoterpenes from isoprene under nickel/iodine cascade catalysis.

monoterpenoids (C10). (2) Regioselectivity. Owing to the existence of four electronically unbiased alkenyl carbons on isoprene, >90 cyclic nucleophilic dimerization isomers of isoprene could be yielded theoretically[23]. (3) Redoxselectivity. In order to realize the nucleophilic aromatization of cyclic monoterpene, up to 4 different hydrogen atoms have to be eliminated regioselectively. It poses high demands on the selectivity of the oxidation system and the compatibility of isoprene dimerization.

By addressing these challenges, we herein demonstrate an efficient nucleophilic aromatization of monoterpenes from isoprene under nickel/iodine cascade catalysis (Fig. 1c). Preliminary mechanistic studies were conducted to understand the cascade catalysis process. Meanwhile, convergent syntheses and orthogonal C–H functionalizations were presented to show the compatibility of this protocol. And

various hybrid terpenyl indoles from the aromatic C10 products were created programmably.

## Results

### Reaction optimization

Unprotected indole **1a** and isoprene **2a** were chosen as model substrates for the investigation of nucleophilic aromatization under nickel/iodine cascade catalysis (Table 1). In the presence of $K_2S_2O_8$, only prenylation products **3a**, **4a** (33% total yield) and diprenylated product **5a** (20% yield) were detected using $Ni(cod)_2$ and $I_2$ as co-catalyst in one step procedure (entry 1). With the addition of IPr·HCl/NaO$^t$Bu, the direct hydroarylation of isoprene was inhibited and no heteroarylative cyclotelomerization was observed (entry 2). When the reaction was carried out with two stepwise procedures in one-pot,

**Table 1 | Optimization of reaction conditions[a]**

1a + 2a

1) Ni(cod)₂/ligand (5.0 mol%), Solvent A, 12 h
2) I₂ (20 mol%), Oxidant, Solvent B, 18 h additives, T °C

Products: 3a, 4a, 5a; 6: from 9a; 7: from 9b (complicated mixture of isomers); C10 6+7 m/z = 253.2; 8a

| Entry | Cat. 1 | Cat. 2 | Additives | Sol. A | Sol. B | 3a + 4a (%) | 5a (%) | 6 + 7 (%) | 8a (%) |
|---|---|---|---|---|---|---|---|---|---|
| 1[b] | Ni(cod)₂ | I₂ | K₂S₂O₈ | THF | | 33 | 20 | – | – |
| 2[b] | Ni(cod)₂/IPr·HCl/NaOtBu | I₂ | K₂S₂O₈ | THF | | 2 | – | – | – |
| 3[c] | Ni(cod)₂/IPr·HCl/NaOtBu | I₂ | K₂S₂O₈ | THF | | 29 | 5 | – | – |
| 4 | Ni(cod)₂/IPr·HCl/NaOtBu | I₂ | K₂S₂O₈ | THF | | – | – | 1 | 28 |
| 5 | Ni(cod)₂/IPr·HCl/NaOtBu | I₂ | K₂S₂O₈ | THF | | 16 | 6 | – | – |
| 6[d] | Ni(cod)₂/dppe | I₂ | K₂S₂O₈ | THF | | – | – | 6 | 3 |
| 7 | Ni(cod)₂/PCy₃ | I₂ | K₂S₂O₈ | THF | | 3 | – | 2 | 1 |
| 8 | Ni(cod)₂/IMes·HCl/NaOtBu | I₂ | K₂S₂O₈ | PhMe | THF | – | – | 3 | 48 |
| 9 | Ni(cod)₂/IPr·HCl/NaOtBu | I₂ | K₂S₂O₈ | n-hexane | THF | – | – | 3 | 51 |
| 10 | Ni(cod)₂/IPr·HCl/NaOtBu | I₂ | K₂S₂O₈ | n-hexane | 1,4-dioxane | – | – | 42 | 13 |
| 11 | Ni(cod)₂/IPr·HCl/NaOtBu | I₂ | DMSO | n-hexane | THF | – | – | – | – |
| 12 | Ni(cod)₂/IPr·HCl/NaOtBu | NIS | K₂S₂O₈ | n-hexane | THF | – | – | – | – |
| 13 | Ni(cod)₂/IPr·HCl/NaOtBu | I₂ | KI/K₂S₂O₈ | n-hexane | THF | – | – | 4 | 58 |
| 14 | Ni(cod)₂/IPr·HCl/NaOtBu | I₂ | KI/Na₂S₂O₈ | n-hexane | THF | – | – | 4 | 64 |
| 15 | Ni(cod)₂/IPr·HCl/NaOtBu | I₂ | KI/AdCO₂H Na₂S₂O₈ | n-hexane | THF | – | – | 7 | 68 |
| 16[e] | Ni(cod)₂/IPr·HCl/NaOtBu | I₂ | KI/(PhO)₂PO₂H Na₂S₂O₈ | n-hexane | THF | – | – | 5 | 82 |

[a]Reaction conditions: Step I: 2a (1.2 mmol), [Ni]/ligand (5.0 mol%), [Ni]/ligand (5.0 mol%), base (10 mol%), solvent A (0.50 mL), 100 °C, 12 h; Step II: 1a (0.20 mmol), I₂ or NIS (20 mol%), solvent B (0.50 mL), 100 °C, 18 h. Yield and selectivity were determined by GC-FID analysis with trimethoxybenzene as the internal standard.
[b]Operated in one-step procedure, 80 °C, 18 h.
[c]80 °C.
[d]Ligand (10 mol%).
[e]24 h for the second step.

aromatic C10 product **8a** (28% yield) accompanied by trace of the other cyclic C10 products **6** and **7** was obtained when the temperature was increased from 80 to 100 °C (entries 3 and 4). Obvious impact on reactivities and selectivities was observed through the evaluation of ligands (entries 5–7). Bisphosphine ligand dppe only promoted the undesirable prenylation (entry 5). And the yield of targeted product **8a** decreased significantly when using PCy$_3$ or less bulky IMes as ligand (entries 6 and 7 *vs* 4). The screening of solvents showed that nonpolar solvents such as toluene (48% yield of **8a**), *n*-hexane (51% yield of **8a**) for the first step brought better results (entries 8 and 9 vs 4). An obvious adverse effect on the formation of **8a** when the solvent for the second step was changed from THF to 1,4-dioxane (entries 10 *vs* 4).

When DMSO was used instead of K$_2$S$_2$O$_8$ as oxidant[47], reaction was completely unproductive (entry 11). The same result was also obtained as I$_2$ was replaced with NIS (entry 12). With the addition of catalytic amount of KI, the yield of **8a** increased from 51% to 58% (entry 13). Besides, when using Na$_2$S$_2$O$_8$ instead of K$_2$S$_2$O$_8$, the yield of desired product **8a** could be further slightly improved (entry 14). Notably, acids showed to be effective additives to facilitate the reaction (entries 15 and 16), and (PhO)$_2$PO$_2$H emerged as the preferred additive for the formation of **8a** (82% yield) in terms of reactivity and selectivity.

## Substrate scope

With the optimized conditions in hand, we subsequently explored the generality of nucleophilic aromatization of monoterpenes from isoprene under nickel/iodine cascade catalysis (Fig. 2a). Subjecting unsubstituted **1a** to the standard conditions furnished aromatic C10-substituted indole **8a** in 79% isolated yield. Methyl and fluorine on the phenyl ring, regardless of their positions, were all well-tolerated (**8b, 8d, 8e, 8n** and **8o**). In addition, indoles with either electron-donating (–OMe) or electron-withdrawing (–CN, –NO$_2$, –CO$_2$Me) substituents all reacted with isoprene **2a** in good reactivities (**8c** and **8j-8m**, 51-94% yields). It is noteworthy that the common leaving groups such as –Cl, –Br, –I and –OTf, which could offer useful handles for further synthetic manipulations, were also compatible with the process to provide the corresponding products (**8f–8i**) in decent yields. Meanwhile, a scale-up experiment under the standard conditions was performed to afford the desired product **8a** in 1.55 g with 78% yield.

For 3-substituted indole substrates, the C10 block could be introduced into the C$^2$-site of indoles (Fig. 2b). Various 3-Me indoles (**1p-1s**) were suitable substrates for the desired cross-coupling (58-81% yields). Notably, this transformation can be further extended to indole possessing ester-substituted alkyl group at C$^3$ position (**1t**). For other nucleophiles, indazoles, benzotriazoles and pyrroles showed acceptable reactivities and selectivities (Fig. 2c). With the enhancement of electron-withdrawing substituents, the reigoselectivities (N$^1$ or N$^2$) of indazoles decreased but the total yields were higher (**8u–8y**). In addition, when using benzotriazole as substrate, the regioselectivity was switched (**8z**, N$^2$/N$^1$ = 3:1). Simple pyrrole **1aa** could be modified by two C10 blocks simultaneously to give functionalized pyrrole **8aa**. Besides, both 2-substituted and 2,3-disubstituted pyrroles (**1bb** and **1cc**) could be successfully applied in this transformation as well.

In absence of the nickel, the targeted products could be observed using limonene instead of isoprene under I$_2$ catalysis. Control experiments further confirmed that I$_2$ and persulfate were indispensable for the process of nucleophilic aromatization (Please see Supplementary Tables 1-4 for details). Next, we sought out to assess the substrate scopes under the optimized condition (Fig. 3a). The coupling of unprotected indole **1a** and limonene **9a** under oxidative condition gave the desired product **8a** in 76% yield. Various indoles bearing –OMe, –F, –Br, and –I groups all reacted with limonene **9a** in good reactivities (**8c, 8d, 8g** and **8h**). Notably, free hydroxyl substituted indole was also applicable (**8dd**) and 5-CN or 6-CO$_2$Me indole showed higher reactivities than the reaction from isoprene (**8j** and **8m**). In

addition, 3-methyl indole and indazole also had good compatibility under this metal-free condition (**8r** and **8u**).

Other natural terpenoids were also applicable in this protocol (Fig. 3b). Interestingly, an unusual cleavage of C−C bond occurred to give the product **8m** when limonene **9a** was replaced with carene, α-pinene or β-pinene (**9c–9e**). And an excellent result was obtained (91% yield of **8m**) when terpinolene, which has a tetrasubstituted C = C bond, was applied in the reaction (**9f**). When K$_2$S$_2$O$_8$ was replaced with DMSO and a mixed solvent was used (THF/MeNO$_2$ = 1:1 *v/v*), the couplings of carvone and dihydrocarvone (**9g** and **9h**) with indole exhibited good outcomes for hydroxyl product **8ee** (71% and 43% yield, respectively). Besides, bisabolene, which is widely used in food flavors, could be converted into the product fitted with C15 block and an additional exocyclic C = C bond was formed simultaneously (**8ff**).

To illustrate the practical utility of this protocol, convergent synthesis was performed from the reaction of methyl indole-6-carboxylate **1m** with mixture of olefins (Fig. 3c). Good yield of **8m** was obtained from the mixture of monoterpenes (limonene, carene, α-pinene, β-pinene and terpinolene, mixed in equal proportions). As an important industrial raw material extracted from coniferous plants, the turpentine mainly composed of α-pinene and β-pinene could afford the convergent product **8m** in 63% yield. Commercially available limonene capsule was also attempted for the reaction and a satisfactory result (71% yield) was achieved. It's worth noting that the peels of lemons and citrus, which are often thought as waste, contain high amounts of limonene and thus have the potential to be used as renewable materials. The distillate collected by a simple azeotropic distillation of lemon peel could be successfully applied in this transformation as well (75% yield).

Meanwhile, the scope of aryl alkenes was investigated under the I$_2$ catalysis without the oxidants (Fig. 3d). The reaction exhibited no obvious loss in both reactivities and selectivities when various (1,1-, 1-, and 1,2-)-substituted alkenes were subjected (**8gg–8jj**). In addition, this strategy can be further extended to 1,3-dienes and 4,3-hydroarylation products (**8kk** and **8ll**) could be obtained with high selectivities. These results also suggest that with the help of I$_2$ and oxidant, an oxidative aromatization of monoterpenes may take place before the nucleophilic addition during the aforementioned nucleophilic aromatization of monoterpenes.

## Mechanistic investigations

In order to facilitate the exploration of mechanism, three sub-reactions (dimerization, oxidative aromatization, hydroarylation) were disassembled to study respectively. For cyclotelomerization, the kinetic experiment was carried out to interpret the process of generation of isoprene dimers under nickel catalysis (Fig. 4a). The reaction steadily produced two [4 + 2] cycloaddition products **9a** and **9b** at the early stage. And quantitative conversion of isoprene was reached after five hours. Notably, no other cyclic or acyclic isomers were observed during this process. For aromatization, three isomerization products (**10a, 10b** and **9f**) and two aromatization products (**11a** and **12a**) were detected during the reaction of limonene **9a** under I$_2$ catalysis (Fig. 4b). At the beginning, limonene **9a** was rapidly consumed to form the aforementioned products. The yield of **9f** was higher than that of the other two isomers (**10a** and **10b**) and they all stayed at a low concentration with the consumption of limonene **9a**. Meanwhile, two aromatization products (**11a** and **12a**) kept increasing until limonene **9a** vanished. These results suggest that terpinolene **9f** is probably the intermediate for the formation of aromatic products **11a** and **12a** through oxidation. Similar studies were conducted to figure out the mechanism when other monoterpenes (carene **9c**, α-pinene **9d** or β-pinene **9e**) replaced limonene **9a** as substrates (Please see Supplementary Figs. 4−6 for details).

For hydroarylation, the generation trend of these four products (**10a, 10b, 9f** and **11a**) is consistent with the process of aromatization,

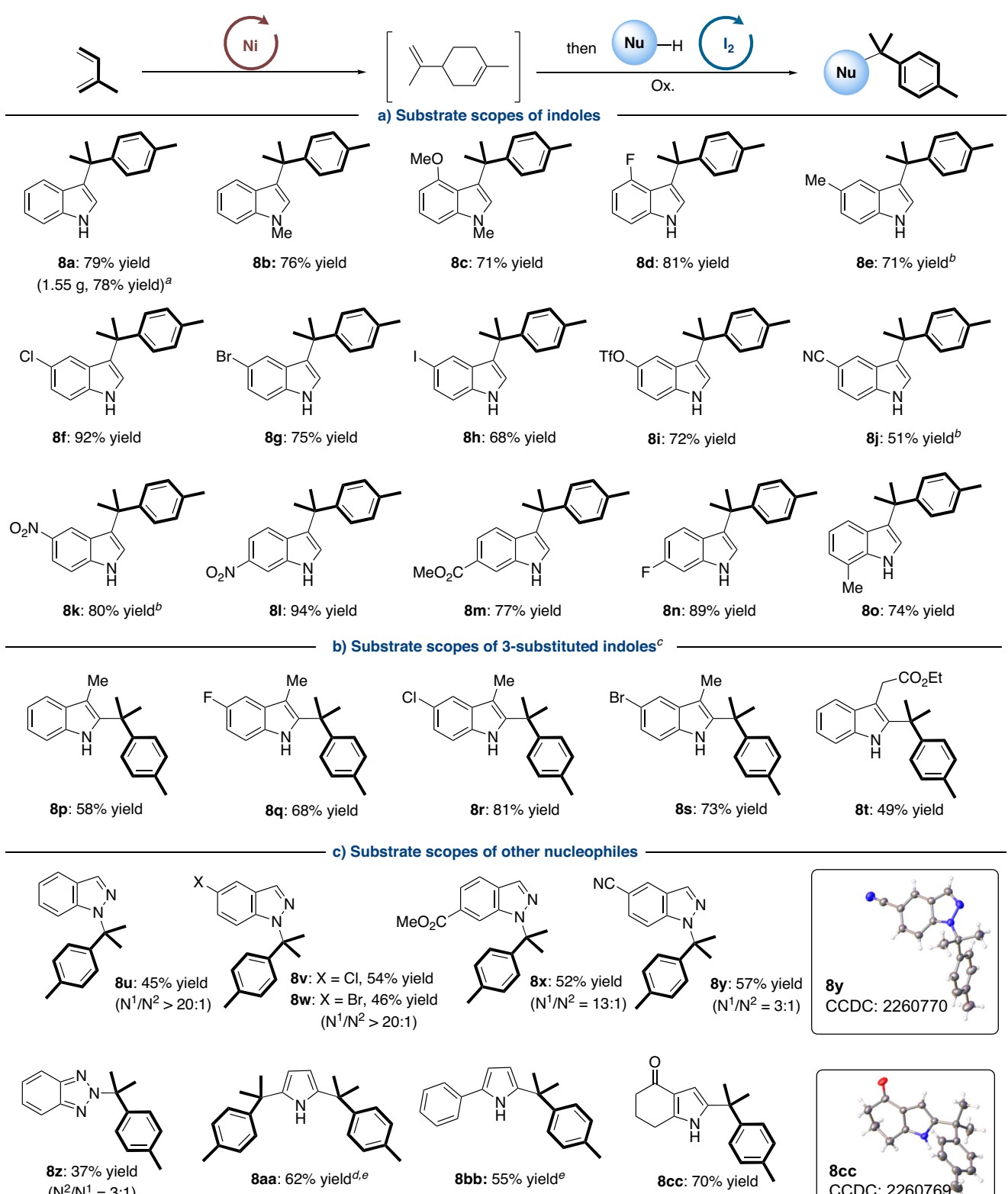

**Fig. 2 | Substrate scopes towards nucleophilic aromatization of isoprene.**
Conditions: Step I: **2a** (1.2 mmol), Ni(cod)₂/IPr·HCl (5.0 mol%), NaOᵗBu (10 mol%), *n*-hexane (0.50 mL), 100 °C, 12 h; Step II: **1** (0.20 mmol), I₂ (20 mol%), KI/(PhO)₂PO₂H (10 mol%), Na₂S₂O₈ (2.0 equiv.), THF (0.50 mL), 100 °C, 24 h. Isolated yields were given in all cases. Regio-selectivities were determined by ¹H NMR analysis. ᵃ**1a** (8.0 mmol), **2a** (48 mmol), *n*-hexane/THF (25 + 25 mL), 48 h for the second step; ᵇ48 h for the second step; ᶜ80 °C for the second step; ᵈ**1aa** (0.10 mmol); ᵉFiltration was required after the first step.

but **12a** maintained at a lower concentration until the target product **8a** no longer increased (Fig. 4c). This indicates that aromatization product **12a** may be the active intermediate. And the formation rate of **12a** is slower in absence of KI (Please see Supplementary Fig. 10 for details). In contrast to the continuous increase of **8a**, undesirable product **6** was produced only at the beginning of the reaction and then remained unchanged nearly. These results indicate the aromatization of cyclic monoterpene occurs before the hydroarylation. When

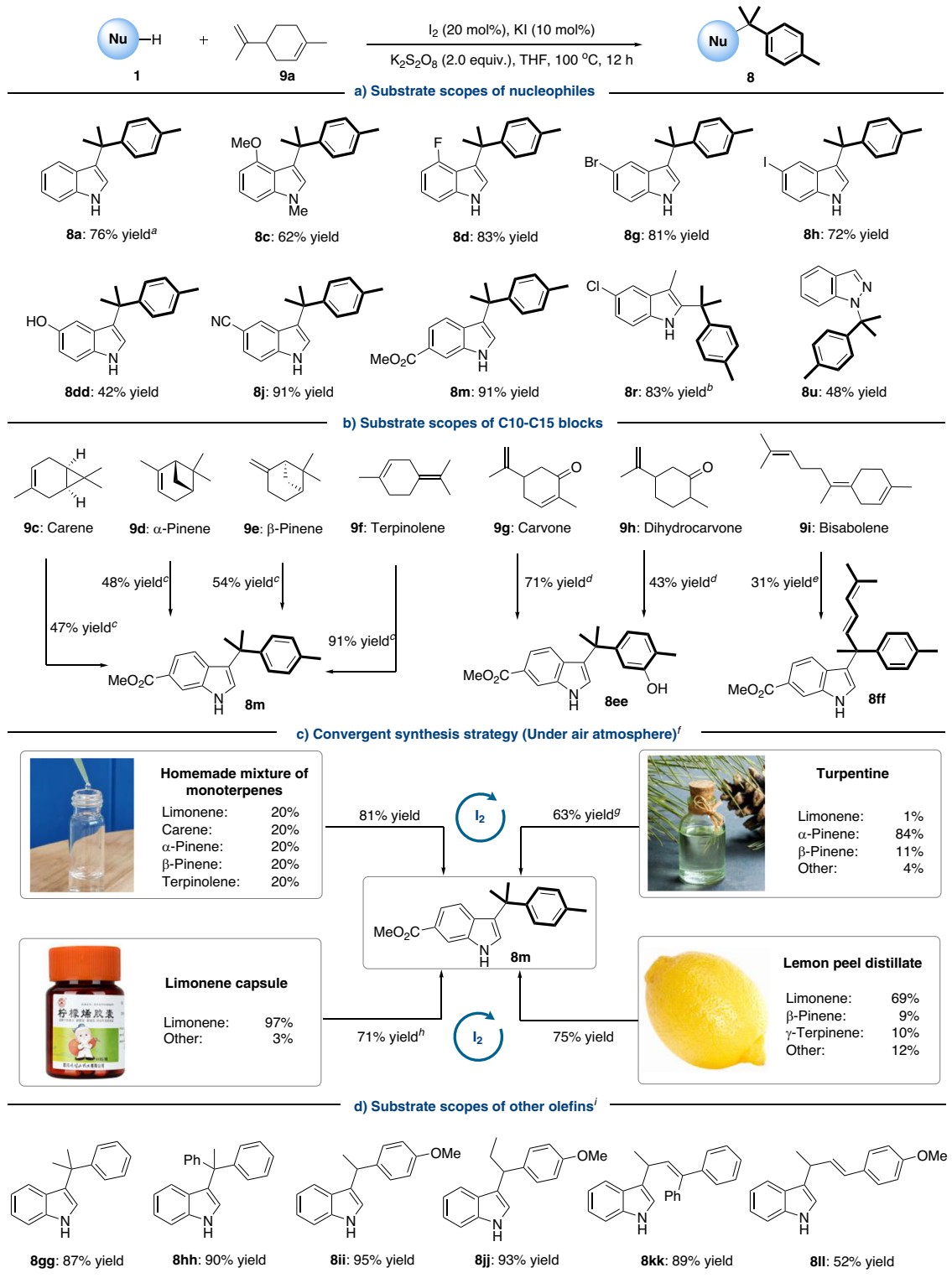

**Fig. 3 | Substrate scopes towards nucleophilic aromatization of olefins.** Conditions: **1** (0.20 mmol), **9** (0.30 mmol), I₂ (20 mol%), KI (10 mol%), K₂S₂O₈ (2.0 equiv.), THF (0.50 mL), 100 °C, 12 h. Isolated yields were given in all cases. [a]6 h; [b]80 °C, 18 h; [c]**9** (0.40 mmol), K₂S₂O₈ (2.5 equiv.), 24 h; [d]DMSO was used instead of

K₂S₂O₈, THF/MeNO₂ (0.50/0.50 mL), 24 h; [e]18 h; [f]The proportion of each component was determined by the ratio of peak area of GC-FID and reactions (mixture of terpenes 54.5 mg) were conducted under the air atmosphere with 18 h; [g]24 h; [h]A grain of limonene capsule; [i]**9** (0.25 mmol) without K₂S₂O₈, 18 h.

1,3-dideuterated indole was treated under the standard condition (Please see Supporting Information for details), both indole ring and aromatic C10 moiety were deuterated partially. The deuterium rate at C² of indole (23% D) was obviously higher than that at other sites, which indicates that C² of indole is more reactive under this protocol.

Therefore, when active C³ position of indole was substituted, the C10 block was easily introduced onto the C² site of indole. Furthermore, the KIE experiment ($k_H/k_D = 1.23$, see Supplementary Fig. 8 for details) suggests that the hydroarylation is probably not the rate-determining step.

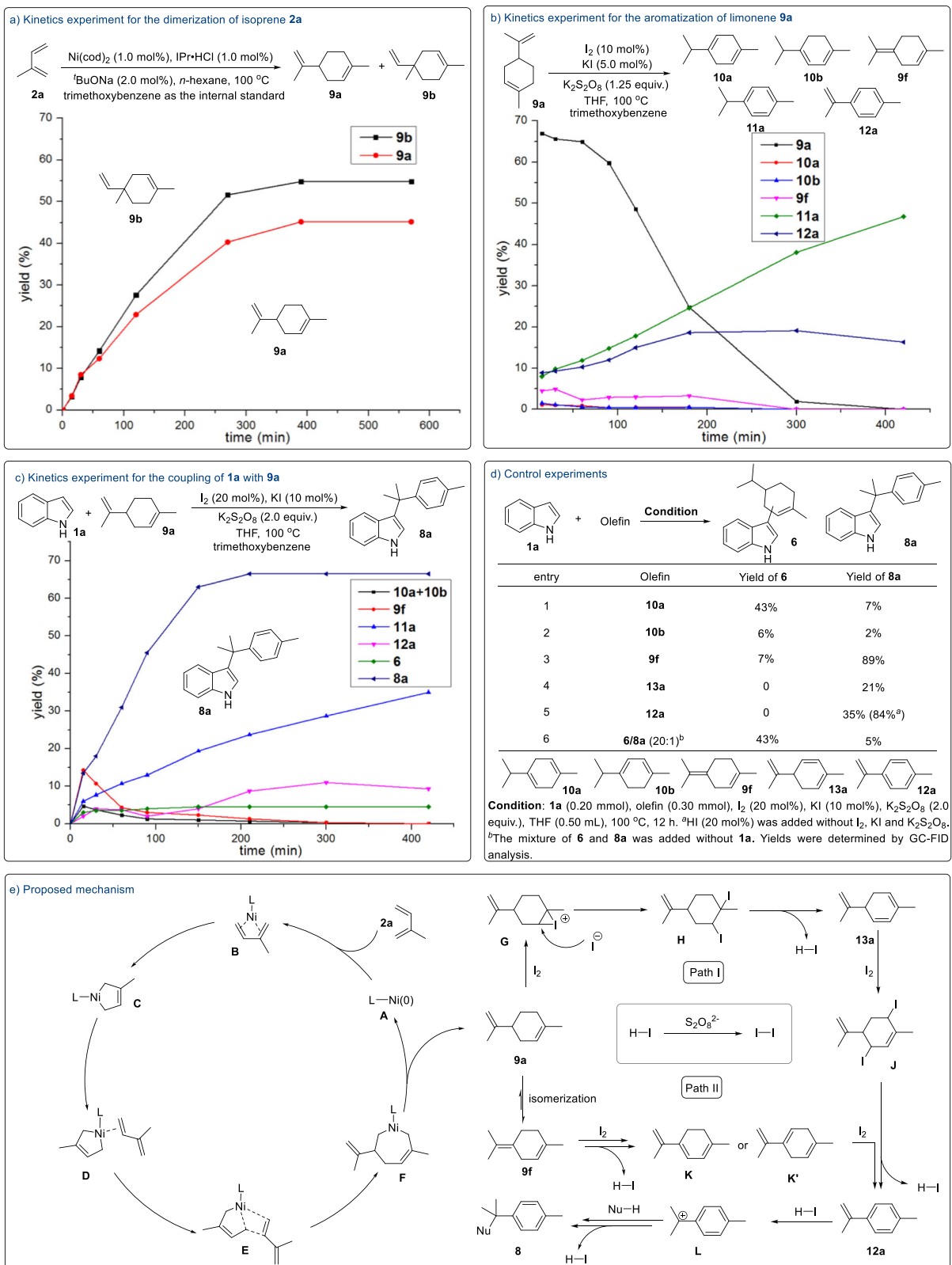

**Fig. 4 | Mechanistic studies and proposed mechanism. a** Kinetics experiment for the dimerization of isoprene **2a**; **b** Kinetics experiment for the aromatization of limonene **9a**; **c** Kinetics experiment for the coupling of **1a** with **9a**; **d** Control experiments; **e** Proposed mechanism.

Additional control experiments were carried out to further elucidate the roles of potential intermediates (Fig. 4d). When the reactions were conducted with substrates only bearing endocyclic double bonds, the side-products **6** rather than the desired product **8a** were generated preferentially (Fig. 4d, entries 1 and 2). In comparison, olefin **9f** which

having exocyclic double bond was more inclined to form the target product **8a** (Fig. 4d, entry 3). These results indicate that the double bond located at the isopropyl unit is the guarantee for the selective formation of **8a**. In addition, triene **13a** could also be transformed into **8a** with 21% yield under the standard condition. Using HI (20 mol%)

instead of the combo of I₂, KI and K₂S₂O₈, a higher yield of product **8a** could be obtained from *p*-cymenene **12a** through a Friedel-Crafts type reaction. Besides, when the mixture of **6** and **8a** was added instead of indole and olefin into the reaction, no conversion of **6** into **8a** was observed (Fig. 4d, entry 6). It ruled out the possibility that hydroarylation occurs before oxidative aromatization of monoterpenoid.

On the basis of the above observations, a plausible mechanism was proposed for the nucleophilic aromatization of monoterpenes from isoprene via nickel and iodine catalysis (Fig. 4e). For the isoprene **2a** dimerization process, active Ni(0) species **A** which is formed in situ from Ni(cod)₂, IPr·HCl and NaOʳBu firstly coordinates with one molecule isoprene **2a** to generate the complex **B**. Subsequently, Ni complex **B** undergoes oxidative cyclometallation to give a five-membered nickelacycle **C**. Through the coordination with another isoprene **2a**, complex **D** proceeds with a migratory insertion to furnish a seven-membered nickelacycle **F** via a transition state **E**. With the formation of new C–C bond, reductive elimination of Ni(II) species **F** yields the limonene **9a** and active Ni(0) species **A**.

There are two possible paths for nucleophilic aromatization (Fig. 4e, right)[48–50]. For path I, iodonium species **G** is initially obtained from limonene **9a** with the help of I₂. Through a nucleophilic attack by iodide, vicinal diiodide compound **H** is generated, which gives the intermediate **13a** smoothly by an elimination of H–I. A subsequent 1,4-addition yields a skipped diiodide species **J**. And olefin **12a** is formed from species **J** via double elimination of H–I. Finally, with the aid of the early released H–I, carbonium ion intermediate **L** is generated and reacts with nucleophile successfully to yield target product **8**. The required I₂ is regenerated from the oxidation of H–I by persulfate. For path II, an alkenyl isomerization **9a** occurs to give more thermodynamic stable diene **9f**. Through repeat processes of di-iodination/HI elimination (**9f-K/K′−12a**), olefin **12a** is then generated as the intermediate for the formation of desired product **8**.

Inspired by parallel kinetic resolution, we envisioned whether it would be possible to develop a parallel transformation of these two regioisomers (**9a** and **9b**) of monoterpene with different nucleophiles in one same pot. As two important but similar heterocycles, benzofuran and indole were chosen as competitors for this proposal. Impressively, an orthogonal C–H monoterpenation of benzofuran and indole occurred smoothly to give product **15a** and **8f** in a chemo-, regio-, and redoxselective fashion. This orthogonal C–H functionalizations also showed good substrate generality (Fig. 5a). Benzofurans bearing electron-donating (-OMe) and -withdrawing (-CO₂Me) groups all underwent the desired cross-coupling smoothly (**15b** and **15c**). It should be noted that the borate group, which could be easily further modified, was well compatible under the current conditions (**15d**). The phenyl group on different sites of phenyl ring had no significant impact on the reactions (**15e** vs **15f**).

## Transformations

The hybrid terpenyl indoles could be used as key building blocks in the total synthesis of a series of natural products. Thus, concise and divergent constructions of various hybrid terpenyl indoles have been demonstrated (Fig. 5b). Treatment of **8a** with a mixture of prenyl bromide and NaH successfully introduced linear C5 block onto the N atom of indole with high efficiency (**16a**). Under palladium catalysis, reverse N-prenylation occurred to furnish the corresponding product **16b** from **8a** and 2-methyl-2-butene[51]. Through the strategy of dearomatization−rearomatization, N-cyclopentylindole **16c** was formed with 57% yield in one step[52]. When using prenyl borates as C5 block, Pd-catalyzed Suzuki reaction underwent smoothly to deliver 5-prenyl indole **16d** in 72% yield[53]. Furthermore, with the help of bases and additives, the linear and cyclic C10 blocks could be introduced into the N atom of indoles (**16e** vs **16f**) respectively[54]. For the another side of the indole ring, diaryl C10 product **16g** could be accessed by a nickel catalyzed carbonyl-Heck reaction from cuminaldehyde and **8i**[55].

Through a traditional Heck reaction, linear C10 compound **16h** substituted at 5-position of indole was finally obtained in 82% yield[56].

## Discussion

In summary, a bioinspired nucleophilic aromatization of monoterpenes from isoprene has been developed under the cascade catalysis of nickel and iodine. Various nucleophiles such as indoles, indazoles, benzotriazole and pyrroles were suitable for this protocol. A series of natural terpenes were transformed to aromatic C10 or C15 blocks successfully by I₂ catalysis. Based on the good tolerance of different monoterpenes, convergent synthesis strategy was implemented smoothly by using drugs and plant extracts. Meanwhile, preliminary mechanistic studies provided evidence for possible catalytic pathways. Firstly, through the regulation of IPr ligand, the cyclodimerization of isoprene proceeds under Ni catalysis. Then, In the presence of K₂S₂O₈, a highly selective nucleophilic addition occurs quickly following the oxidative aromatization of limonene under I₂ catalysis. An orthogonal C–H functionalization of two regioisomers of monoterpene with different nucleophiles was realized in one same pot. Moreover, various hybrid terpenyl indoles were programmatically assembled to illustrate the practical utility of this protocol. Further studies and application on transformations of isoprene are underway in our laboratory.

## Methods

### General procedure for Ni/I₂ catalyzed nucleophilic aromatization from isoprene

Step I: In a glove box, a sealed tube was charged with Ni(cod)₂ (0.01 mmol, 5.0 mol%), IPr·HCl (0.01 mmol, 5.0 mol%), NaOʳBu (0.02 mmol, 10 mol%), isoprene **2a** (1.2 mmol), *n*-hexane (0.50 mL) at room temperature. The reaction tube was sealed with a Teflon screw cap and removed from the glove box. Then, the reaction mixture was stirred at 100 °C for 12 hours. Step II: As the reaction mixture was cooled to room temperature, (PhO)₂PO₂H (0.02 mmol, 10 mol%) was added to the reaction tube and stirred for 5 minutes. Then KI (0.02 mmol, 10 mol%), K₂S₂O₈ (0.40 mmol, 2.0 equiv.), nucleophile **1** (0.20 mmol), I₂ (0.04 mmol, 20 mol%) and THF (0.50 mL) were added into the reaction mixture and stirred at 100 °C for additional 24 h. Direct purification by column chromatography on silica gel using petroleum ether and ethyl acetate afforded the corresponding product **8**.

### General procedure for I₂ catalyzed nucleophilic aromatization from terpenes

In a glove box, a sealed tube was charged with KI (0.02 mmol, 10 mol%), K₂S₂O₈ (0.40 mmol, 2.0 equiv.), nucleophile **1** (0.20 mmol), I₂ (0.04 mmol, 20 mol%), **9** (0.30 mmol) and THF (0.50 mL) at room temperature. The reaction tube was sealed with a Teflon screw cap and removed from the glove box. Then, the reaction mixture was stirred at 100 °C for 12 h. Direct purification by column chromatography on silica gel using petroleum ether and ethyl acetate afforded the corresponding product **8**.

### General procedure for I₂ catalyzed hydroarylation of olefins

In a glove box, a sealed tube was charged with KI (0.02 mmol, 10 mol%), indole **1a** (0.20 mmol), I₂ (0.04 mmol, 20 mol%), **9** (0.25 mmol) and THF (0.50 mL) at room temperature. The reaction tube was sealed with a Teflon screw cap and removed from the glove box. Then, the reaction mixture was stirred at 100 °C for 18 h. Direct purification by column chromatography on silica gel using petroleum ether and ethyl acetate afforded the corresponding product **8**.

### General procedure for Orthogonal C–H functionalizations

Step I: In a glove box, a sealed tube was charged with Ni(cod)₂ (0.01 mmol, 5.0 mol%), IPr·HCl (0.01 mmol, 5.0 mol%), NaOʳBu

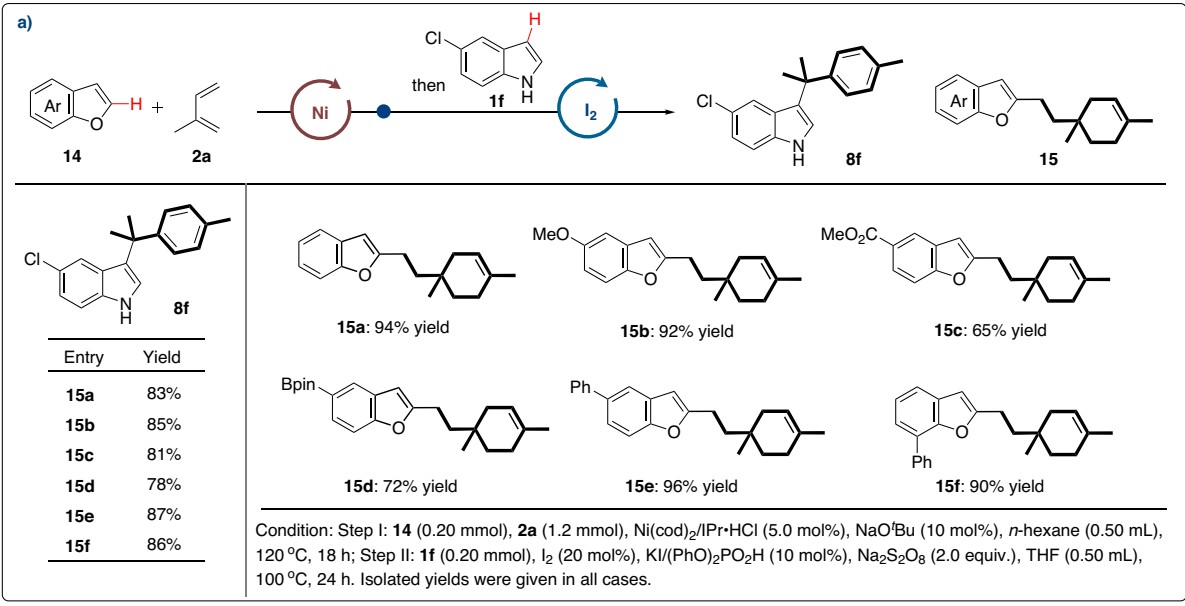

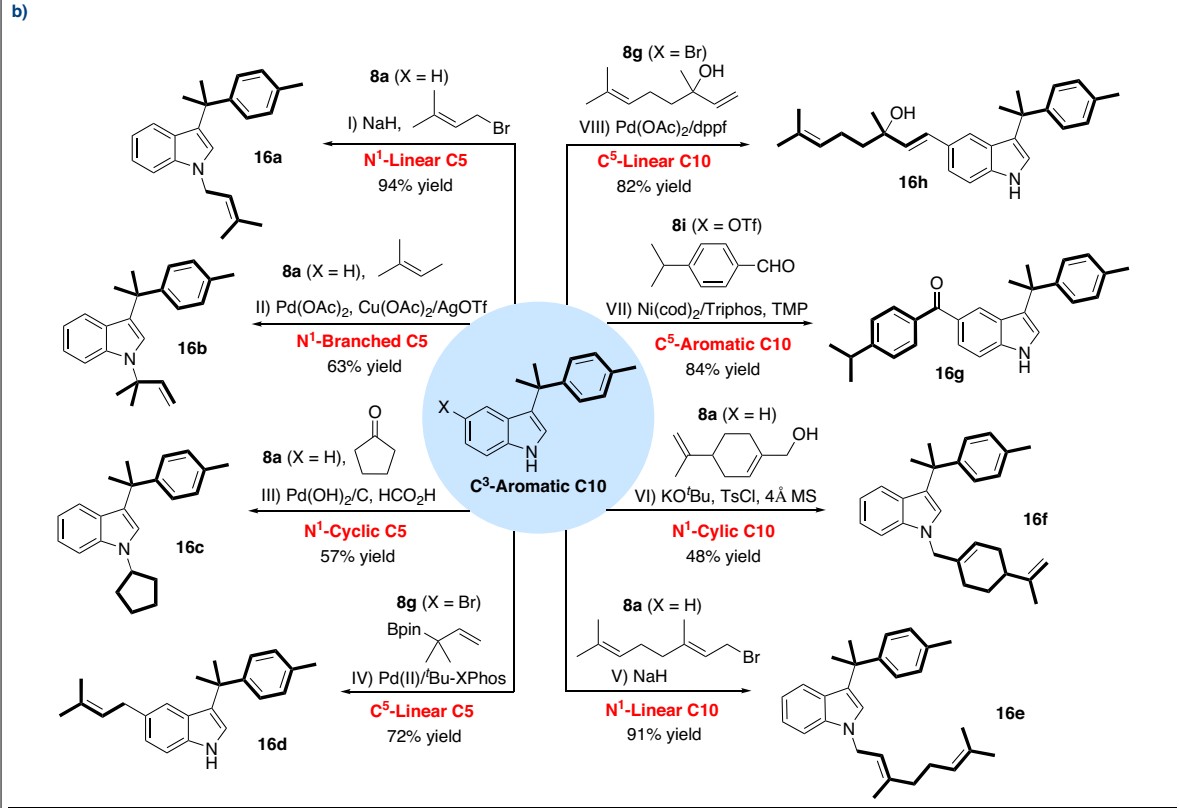

**Fig. 5 | Synthetic applications. a** Orthogonal C–H functionalizations; **b** Concise and programmable construction of hybrid terpenyl indoles.

(0.02 mmol, 10 mol%), benzofuran **14** (0.20 mmol), isoprene **2a** (1.2 mmol), n-hexane (0.50 mL) at room temperature. The reaction tube was sealed with a Teflon screw cap and removed from the glove box. Then, the reaction mixture was stirred at 120 °C for 18 h. Step II: As

the reaction mixture was cooled to room temperature, (PhO)$_2$PO$_2$H (0.02 mmol, 10 mol%) was added to the reaction tube and stirred for 5 min. Then, KI (0.02 mmol, 10 mol%), K$_2$S$_2$O$_8$ (0.40 mmol, 2.0 equiv.), indole **1f** (0.20 mmol), I$_2$ (0.04 mmol, 20 mol%) and THF (0.50 mL)

were added into the reaction mixture and stirred at 100 °C for additional 24 h. Direct purification by column chromatography on silica gel using petroleum ether and ethyl acetate afforded the corresponding products **8f** and **15**.

## Data availability

Crystallographic data for the structures reported in this Article have been deposited at the Cambridge Crystallographic Data Centre, under deposition numbers CCDC 2260769 (**8cc**) and 2260770 (**8y**). Copies of the data can be obtained free of charge via https://www.ccdc.cam.ac.uk/structures/. Data relating to the characterization data of materials and products, general methods, optimization studies, experimental procedures, mechanistic studies and NMR spectra are available in the Supplementary Information. All data are also available from the corresponding author upon request.

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

## Acknowledgements

Financial support from Dalian Outstanding Young Scientific Talent (2020RJ05), and the National Natural Science Foundation of China (22071239) is acknowledged.

## Author contributions

Q.-A.C. conceived and supervised the project. Q.-A.C. and W.-S.Z. designed the experiments. W.-S.Z., D.-W.J., Y.Y., T.-T.S, G.Z., and X.-Y.W. performed the experiments and analyzed the data. All authors discussed the results and commented on the manuscript.

## Competing interests

The authors declare no competing interests.
