## [Peer Review File · Nature Communications]

Nucleophilic Aromatization of Monoterpenes from Isoprene under Nickel/Iodine Cascade CatalysisReviewers' Comments:

Reviewer #1:

Remarks to the Author:

Qing-An Chen and co-workers report a method to combine two isoprene units to ultimately make an aryl group, which is a very nice one-pot transformation. The manuscript's introduction is diffuse, meaning it does not really get to the point of the manuscript. It needs to be focused on the topic of the actual science in the manuscript. The cyclotramerization of isoprene is known, even by this team in their *Nat. Catalysis* paper, so that part is not new. The new part would then be the reaction of limonene with the heterocycles.

Table 1 shows the development and optimization of the reaction, where KI, (PhO)₂PO₂H, Na₂S₂O₈ (oxidant) and cat. I₂ and Ni(NHC) are used. The Ni catalyzed the condensation of two isoprene units to make limonene with and its isomer. The I₂ under oxidative conditions promotes the electrophilic aromatic substitution. These are nicely shown in independent experiments. As shown in Fig 2, many indoles can react at the 3- or (if the 3-position is blocked) at the 2-position. Some other electron rich heterocyclic groups work too and give fine yields of the expected products.

As shown in figure 3, limonene under I₂ catalysis can be used in place of isoprene/Ni, in which case no Ni catalyst is needed. Other derivatives, carene, α -pinene or β -pinene, also work. The chemistry in Fig 3B is surprising to me, and it is interesting. I wonder what the mechanisms of these reactions are.

In Figure 4 they propose a mechanism, which seems very reasonable, although I would expect iterative eliminations to take place rather than formation of a tri-iodide intermediate. Scheme 4d, entry 5 is not clear what the exact difference between the two conditions are.

Figure 5a is basically the same as in this paper, and I am surprised that this manuscript from the same team on a similar reaction was not cited in the text. *Chinese Journal of Catalysis* Volume 49, June 2023, Pages 123-131. I suspect it was published before this manuscript was submitted.

The derivatization of the products is shown in Scheme 6.

The SI seems of high quality. Compounds are fully characterized.

The strengths of the manuscript are some interesting reactivity in the generation of limonene (although it is known with low yields for a long time with many examples, like *Journal of the American Chemical Society*, 1990, vol. 112, # 3, p. 1292 – 1294 and including in the authors' recent work *Nat. Catal.* paper). The downside of this work is that the group added to the indoles and other heterocycles is very specific. Also, there is some overlap with the authors' past work. Most importantly, the authors have not adequately described the utility of the products. Thus, I would put this work at the *Adv. Syn. & Catalysis* level.

Reviewer #2:

Remarks to the Author:

In this work Prof. Chen and co-workers described a nickel/iodine catalyzed nucleophilic aromatization of monoterpenes using isoprene as the cheap and readily available materials. The reaction two isoprenes formed the key C₁₀ intermediate which could react with various nucleophiles to generate various functionalized products. This work could provide an efficient approach for the construction of

unnatural monoterpenoids. The reaction mechanism was systematic investigated and it is reasonable. I recommend publication of this work in Nature Communications after revision.

- 1) First of all, what is the role of KI since iodine was acted as the key promoter in this work?
- 2) Is it possible the nucleophile reaction with one equiv. of isoprene first and then cyclized with another equiv. of isoprene?
- 3) How about the reaction with only two equiv. of isoprenes?
- 4) Besides indoles, how about other electron-ring heteroarenes such as pyrroles??
- 5) Since there are excess of isoprene existed, did the authors analysis byproducts in the reaction mixture?

Reviewer #3:

Remarks to the Author:

In this manuscript, the authors reported a novel nucleophilic aromatization of monoterpenes from isoprene by the cascade catalysis of nickel and iodine, providing a convenient access to monoterpenoids. Various nucleophiles such as indoles, indazoles, benzotriazole and pyrroles were suitable for this protocol. Delicate control experiments were carried out to account for the reaction mechanism and a reasonable mechanism was proposed. The compounds were well-characterized, and the paper was well-written. This reviewer suggests its acceptance after the following issues being addressed.

1. Could iodide intermediate be isolated or detected? Add references for mechanism part.
2. Line 103, 9e should be 9f.
3. Line 171, "9a an 9b" should be "9a and 9b".

Response to reviewer 1:

This reviewer thought this was a very nice one-pot transformation and provided many helpful comments. Thanks for the reviewer's careful reading and constructive suggestions, and we respond to below.

Comments: (1) Qing-An Chen and co-workers report a method to combine two isoprene units to ultimately make an aryl group, which is a very nice one-pot transformation. The manuscript's introduction is diffuse, meaning it does not really get to the point of the manuscript. It needs to be focused on the topic of the actual science in the manuscript.

Response: We appreciate the reviewer's suggestion. In this paper, we firstly introduced the importance of monoterpenoids and its construction challenges. And isoprene is the most cost-effective choice for biomimetic constructions of monoterpenoids. Although there are some reports on the catalytic synthesis of acyclic and cyclic monoterpenoids, precise functionalization of aromatic C10 blocks from isoprene was unexplored. Finally, by regulating the complicated selectivities (chemo-, regio- and redoxselectivity), an efficient nucleophilic aromatization of monoterpenes from isoprene was demonstrated under nickel/iodine cascade catalysis. According to the reviewer's suggestion, we slightly simplify the Fig. 1 to make it easier to understand.

Comments: (2) The cyclotelomerization of isoprene is known, even by this team in their Nat. Catalysis paper, so that part is not new. The new part would then be the reaction of limonene with the heterocycles.

Response: We appreciate the reviewer's suggestion. Generally, one-pot transformation is not a simple combination with two synthetic processes. For current work, it poses high demands on the compatibility between isoprene dimerization and nucleophilic aromatization. Furthermore, the dimerization of isoprene will form monoterpenes (e.g. **9a** and **9b**) which share very strong structural similarities. Therefore, it is a great challenge to develop a catalytic protocol to selectively react one monoterpene over another. In our previous work (ref 46: *Nat. Catal.* **2022**, 5, 708), we realized a Ni-catalyzed asymmetric heteroarylate cyclotelomerization of isoprene *via* the intermediate **9b**. Herein, we developed a nucleophilic aromatization of monoterpenes (*via* limonene **9a**) from isoprene under nickel/iodine cascade catalysis. These two works are very different in terms of catalytic system, regioselectivity, redoxselectivity and substrate scope.

Comments: (3) As shown in figure 3, limonene under I₂ catalysis can be used in place of isoprene/Ni, in which case no Ni catalyst is needed. Other derivatives, carene, α -pinene or β -pinene, also work. The chemistry in Fig 3B is surprising to me, and it is interesting. I wonder what the mechanisms of these reactions are.

Response: Thanks for your constructive suggestion. In order to explore the reaction mechanism of carene, α -pinene and β -pinene, three parallel kinetics experiments were conducted respectively (Please see SI, pages S17 and S18). For the reaction of carene, limonene **9a**, aromatic product **11a** and key intermediate **12a** could be detected during the reaction process. For the reaction of α -pinene and β -pinene, in addition to products **9a**, **11a** and **12a**, the isomerization products **10a**, **10b** and **9f** could be observed.

On the basis of the above observations and previous work on the isomerization of pinene (*J. Am. Oil Chem. Soc.* **2005**, 82, 531.), plausible mechanisms were proposed (Please see SI, page S19). For carene, iodonium species **A'** is initially obtained from carene **9c** with the help of I_2 . Through a nucleophilic attack by iodide, vicinal diiodide compound **B'** is generated, which gives the intermediate **C'** smoothly by elimination of H-I . With the iodonium species **D'** generation, a subsequent electron transfer yields a carbocation **E'**, which could be attacked by iodide ion to form species **F'**. Finally, olefin **12a** is formed from species **F'** via double elimination of H-I . The required I_2 is regenerated from the oxidation of H-I by persulfate.

Through the similar process of ring opening and di-iodination/ HI elimination, olefin **12a** could be obtained from α -pinene **9d** and β -pinene **9e** as well.

The above results and corresponding comments have been added to page S18 of SI and page 8, paragraph 3 of the manuscript.

Comments: (4) Scheme 4d, entry 5 is not clear what the exact difference between the two conditions are.

Response: We appreciate the reviewer's suggestion and have revised the footnote. When a catalytic amount of HI was added instead of I₂, KI and K₂S₂O₈, a higher yield of target product **8a** was obtained from olefin **12a**.

Comments: (5) Figure 5a is basically the same as in this paper, and I am surprised that this manuscript from the same team on a similar reaction was not cited in the text. Chinese Journal of Catalysis Volume 49, June 2023, Pages 123-131. I suspect it was published before this manuscript was submitted.

Response: We appreciate the reviewer's suggestion. Because the number of citing references was restricted when this manuscript was initially submitted to *Nat. Catal.*, only more relevant references were cited. Lately, this manuscript was directed transferred to *Nat. Commun.* Besides, there are obvious differences between the work of Fig. 5a and our previous paper (*Chin. J. Catal.* **2023**, *49*, 123.). In previous paper, C5 or C10 block was introduced onto the same heterocycles (benzimidazoles) depending on the regulation of ligands. However, this work developed a parallel transformation of two regioisomers (**9a** and **9b**) of monoterpene with different nucleophiles (benzofurans and indoles) in one pot. We have added this reference as ref 35.

Comments: (6) The strengths of the manuscript are some interesting reactivity in the generation of limonene (although it is known with low yields for a long time with many examples, like Journal of the American Chemical Society, 1990, vol. 112, # 3, p. 1292 – 1294 and including in the authors' recent work Nat. Cat. paper). The downside of this work is that the group added to the indoles and other heterocycles is very specific. Most importantly, the authors have not adequately described the utility of the products. Also, there is some overlap with the authors past work. Most importantly, the authors have not adequately described the utility of the products.

Response: We appreciate the reviewer's comments. The previous work (*J. Am. Chem. Soc.* **1990**, *112*, 1292.) has been cited as ref 45. As this reference showed, different dimers of isoprene mixed together with low yields. It is of great challenge to selectively form specific isoprene dimer and couple with nucleophiles. In our previous work (ref 46: *Nat. Catal.* **2022**, *5*, 708), we realized a Ni-catalyzed asymmetric heteroarylation cyclotramerization of isoprene *via* the intermediate **9b**. Herein, we developed a nucleophilic aromatization of monoterpenes (*via* limonene **9a**) from isoprene under nickel/iodine cascade catalysis. These two works are very different in terms of catalytic system, regioselectivity, redoxselectivity and substrate scope.

Besides indoles, indazoles, benzotriazole and pyrroles were also suitable for this protocol (Please see Fig. 2c). Then, it is noted that other C10 (**9c-9h**) or C15 (**9i**) blocks could be also coupled with heterocycles (Please see Fig. 3b). Meanwhile, the reaction exhibited no obvious loss in both reactivities and selectivities when various (1,1-, 1-, and 1,2-)-substituted alkenes were subjected (Please see Fig. 3d).

In nature, various indole alkaloids were equipped with different terpenyl groups (Please see ref 1). And this work provided a complementary approach for the creation of unnatural monoterpenoids

(Figs. 2 and 3). To illustrate the practical utility of this protocol, convergent synthesis was performed from the reaction of methyl indole-6-carboxylate **1m** with mixture of monoterpenes (Fig. 3c). Besides, concise and divergent constructions of various hybrid terpenyl indoles have been demonstrated in this paper (Fig. 5b), which could be used as key building blocks for further transformations. Furthermore, an orthogonal C–H functionalization of two regioisomers of monoterpene with different nucleophiles was also realized in one same pot (Fig. 5a).

Response to reviewer 2:

This reviewer recommended publication in *Nature Communications* after some revisions. We gratefully acknowledge the reviewer's appreciation on our work and we respond to below.

Comments: (1) What is the role of KI since iodine was acted as the key promoter in this work.

Response: We appreciate the reviewer's suggestion. The kinetics experiment for the coupling of **1a** with **9a** (Please see page 9, Fig. 4c) has indicated that the hydroarylation is a fast process and the key step is the generation of olefin **12a**. To understand the role of KI, the control experiment for the aromatization of limonene was performed (Please see SI, page S25). Obviously, the formation rate of **12a** in absence of KI was slower than that with KI. This result has been added page 8, the last paragraph of the manuscript.

Meanwhile, based on the proposed mechanism (Please see page 9, Fig. 4e), intermediate **H** or **L** was formed from iodonium ion (**G** or **K**) and iodide ion. Thus, the addition of KI probably facilitates the generation of **H** or **L** or other related intermediates. These results and corresponding comments have been added in page S26 of SI.

Comments: (2) Is it possible the nucleophile reaction with one equiv. of isoprene first and then cyclized with another equiv. of isoprene?

Response: Thanks for your constructive suggestion. According to the ref 19 (*Angew. Chem. Int. Ed.* **2019**, *58*, 5438.), reverse-prenylated indole was synthesized. However, the target product **8a** could not be detected when reverse-prenylated indole reacted with isoprene under nickel/iodine catalysis (Please see SI, page S26). Moreover, according to the mechanistic studies of ref 33 (*Nat. Catal.* **2022**, *5*, 708), it is more reasonable that isoprene dimerization occurs before hydroheteroarylation.

Comments: (3) How about the reaction with only two equiv. of isoprenes?

Response: We appreciate the reviewer's suggestion. With the addition of only two equiv. of isoprene addition, 33% yield of **8a** was obtained and no any prenylated indole was detected. The observed low yield of the target product **8a** partially resulted from the formation of undesired dimer **9b** (Please see SI, page S27, Table 11).

Entry	Isoprene	Yield of 9b (%)	Yield of 11a (%)	Yield of 12a (%)	Yield of 6 (%)	Yield of 8a (%)
1	2.0 equiv.	58	8	--	1	33

^aCondition: Step I: **2a** (0.4 mmol), Ni(cod)₂/IPr·HCl (5.0 mol%), NaO^tBu (10 mol%), hexane (0.50 mL), 100 °C, 12 h; Step II: **1a** (0.20 mmol), I₂ (20 mol%), K₂S₂O₈ (2.0 equiv.), KI (10 mol%), THF (0.50 mL), 100 °C, 24 h.

Comments: (4) Besides indoles, how about other electron-ring heteroarenes such as pyrroles?

Response: We appreciate the reviewer's suggestion. Besides indoles, indazoles, benzotriazole and pyrroles were also suitable for this protocol (Please see the page 5, Fig. 2c)

Comments: (5) Since there are excess of isoprene existed, did the authors analysis byproducts in the reaction mixture?

Response: We appreciate the reviewer's suggestion. The reaction mixture was analyzed by GC-FID and the excess isoprene formed dimer **9b** and aromatization product **11a** (Please see SI, page S27, Table 11).

Entry	Isoprene	Yield of 9b (%)	Yield of 11a (%)	Yield of 12a (%)	Yield of 6 (%)	Yield of 8a (%)
2	6.0 equiv.	54	7	--	5	79

^aCondition: Step I: **2a** (1.2 mmol), Ni(cod)₂/IPr·HCl (5.0 mol%), NaO^tBu (10 mol%), hexane (0.50 mL), 100 °C, 12 h; Step II: **1a** (0.20 mmol), I₂ (20 mol%), K₂S₂O₈ (2.0 equiv.), KI (10 mol%), THF (0.50 mL), 100 °C, 24 h. Yields were determined by GC-FID.

Response to reviewer 3:

This reviewer recommended publication in *Nature Communications* after some revisions. We gratitude the reviewer's appreciation on our work and we respond to below.

Comments: (1) Could iodide intermediate be isolated or detected? Add references for mechanism part.

Response: Thanks for your constructive suggestion. Indeed, it would be more convincing if the iodide intermediate was isolated or detected. However, only iodinated aromatization product was detected by HRMS (Please see SI, pages S26 and S27). In several previous works (*J. Chem. Soc., Chem. Commun.*, **1987**, 1491; *Tetrahedron Lett.* **2015**, 56, 7197; *Biomacromolecules* **2021**, 22, 514.), some diiodide compounds have been synthesized or detected from olefins with the help of I₂, which were unstable and could be further transformed easily. And these references have been added as refs 48-50.

Comments: (2) Line 103, 9e should be 9f.

Response: We appreciate the reviewer's suggestion and have revised "9e" to "9f" (Please see page 6, paragraph 4).

Comments: (3) Line 171, "9a an 9b" should be "9a and 9b".

Response: We appreciate the reviewer's suggestion. The "9a an 9b" has been revised as "9a and 9b" (Please see page 10, last paragraph).

Reviewers' Comments:

Reviewer #1:

Remarks to the Author:

This is the second review of the manuscript. The authors answered some of my concerns.

Rereading the manuscript, there are 6 equiv. isoprene are used (Table 1), but Figure 2 says 2 equiv in the figure caption. This should be fixed. Also, it is not clear at first glance what the dot between the arrows is in this figure. Why not include the actual structures.

While I have stated that the chemistry is mechanistically interesting, I am still unconvinced that it is synthetically useful because of the limited scope of aryl groups that can be introduced with this method (Fig 2 and 3a).

L in the mechanism still seems rather unlikely.

Otherwise, if the other reviewers are satisfied and supportive, publication after minor revisions is fine with me.

Reviewer #2:

Remarks to the Author:

After carefully checking the revised manuscript, I found that the authors have addressed most of issues I made as well as other reviewers made. The explanation is reasonable and proper. I recommend publication of this work in its current formation.

Reviewer #3:

Remarks to the Author:

The manuscript has been well corrected. It can be accepted.

Response to reviewer 1:

This reviewer recommended publication in *Nature Communications* after some minor revisions during the second round of review of the manuscript. Thanks for the reviewer's careful reading and constructive suggestions, and we respond to below.

Comments: (1) there are 6 equiv. isoprene are used (Table 1), but Figure 2 says 2 equiv in the figure caption. This should be fixed. Also, it is not clear at first glance what the dot between the arrows is in this figure. Why not include the actual structures.

Response: We appreciate the reviewer's suggestion. Actually, the amount of isoprene has been noted below the Figure 2 and the number "2" in front of the isoprene was added to show the process of dimerization in the first step. For better understanding, we have removed the number "2" and added the structure of limonene instead of the dot.

Comments: (2) While I have stated that the chemistry is mechanistically interesting, I am still unconvinced that it is synthetically useful because of the limited scope of aryl groups that can be introduced with this method (Fig 2 and 3a).

Response: We appreciate the reviewer's suggestion. It should be noted that other C10 (**9c-9h**) or C15 (**9i**) blocks could be also coupled with heterocycles (Please see Fig. 3b). Meanwhile, the reaction exhibited no obvious loss in both reactivities and selectivities when various (1,1-, 1-, and 1,2-)substituted alkenes and 1,3-dienes were subjected (Please see Fig. 3d). Thus, this method could introduce different blocks into heterocycles from alkenes and was not limited to what was shown in Figures 2 and 3a.

Comments: (3) L in the mechanism still seems rather unlikely.

Response: Thanks for your constructive suggestion. We have revised the corresponding proposed mechanism slightly to make it more clearly (Fig. 4e, **9f** to **12**). The debatable previous intermediate **L** has been removed from the figure. And the statement "Through repeat processes of di-iodination/HI elimination" was corrected as "Through repeat processes of di-iodination/HI elimination (**9f-K/K'-12a**)" in the manuscript.

Response to reviewer 2:

This reviewer recommended publication of this work in its current formation.

Response: We appreciate the reviewer's suggestion and help in the first round of review.

Response to reviewer 3:

This reviewer recommended that this article can be accepted by *Nature Communications*.

Response: We appreciate the reviewer's suggestion and help in the first round of review.